# Orthopaedic Implant-Associated Staphylococcal Infections: A Critical Reappraisal of Unmet Clinical Needs Associated with the Implementation of the Best Antibiotic Choice

**DOI:** 10.3390/antibiotics11030406

**Published:** 2022-03-17

**Authors:** Milo Gatti, Simona Barnini, Fabio Guarracino, Eva Maria Parisio, Michele Spinicci, Bruno Viaggi, Sara D’Arienzo, Silvia Forni, Angelo Galano, Fabrizio Gemmi

**Affiliations:** 1Department of Medical and Surgical Sciences, Alma Mater Studiorum University of Bologna, 40138 Bologna, Italy; milo.gatti2@unibo.it; 2SSD Clinical Pharmacology, Department for Integrated Infectious Risk Management, IRCCS Azienda Ospedaliero-Universitaria di Bologna, 40138 Bologna, Italy; 3Bacteriology Unit, Azienda Ospedaliero Universitaria Pisana, 56126 Pisa, Italy; s.barnini@ao-pisa.toscana.it; 4Department of Anaesthesia and Critical Care Medicine, Azienda Ospedaliero Universitaria Pisana, 56126 Pisa, Italy; f.guarracino@ao-pisa.toscana.it; 5UOSD Microbiologia Arezzo PO San Donato, Azienda Usl Toscana Sud Est, 52100 Arezzo, Italy; evamaria.parisio@uslsudest.toscana.it; 6Department of Experimental and Clinical Medicine, University of Florence, 50134 Florence, Italy; michele.spinicci@unifi.it; 7Infectious and Tropical Diseases Unit, Careggi University Hospital, 50134 Florence, Italy; 8Neurointensive Care Unit, Department of Anesthesiology, Careggi University Hospital, 50134 Florence, Italy; bruno.viaggi@gmail.com; 9Agenzia Regionale di Sanità della Toscana, 50141 Florence, Italy; sara_88fi@hotmail.it (S.D.); silvia.forni@ars.toscana.it (S.F.); 10SOD Microbiologia e Virologia, Careggi University Hospital, 50134 Florence, Italy; galanoa@aou-careggi.toscana.it

**Keywords:** orthopaedic implant-associated infections, anti-staphylococcal agents, biofilm, antibiotic bone penetration, outpatient management, long-term safety

## Abstract

Infections associated with orthopaedic implants represent a major health concern characterized by a remarkable incidence of morbidity and mortality. The wide variety of clinical scenarios encountered in the heterogeneous world of infections associated with orthopaedic implants makes the implementation of an optimal and standardized antimicrobial treatment challenging. Antibiotic bone penetration, anti-biofilm activity, long-term safety, and drug choice/dosage regimens favouring outpatient management (i.e., long-acting or oral agents) play a major role in regards to the chronic evolution of these infections. The aim of this multidisciplinary opinion article is to summarize evidence supporting the use of the different anti-staphylococcal agents in terms of microbiological and pharmacological optimization according to bone penetration, anti-biofilm activity, long-term safety, and feasibility for outpatient regimens, and to provide a useful guide for clinicians in the management of patients affected by staphylococcal infections associated with orthopaedic implants Novel long-acting lipoglycopeptides, and particularly dalbavancin, alone or in combination with rifampicin, could represent the best antibiotic choice according to real-world evidence and pharmacokinetic/pharmacodynamic properties. The implementation of a multidisciplinary taskforce and close cooperation between microbiologists and clinicians is crucial for providing the best care in this scenario.

## 1. Introduction

Infections associated with orthopaedic implants represent a major health concern characterized by a remarkable incidence of morbidity and mortality [1,2]. The growth of elderly and immunocompromised populations, coupled with the great efficacy of joint replacement interventions in improving patients’ quality of life, have recently led to an increased incidence of infections associated with orthopaedic implants [3,4]. While in patients undergoing primary joint replacement, the infection rate during the first two years is approximatively less than 1–2%, the infections rate after revision surgery may rise up to 40%. Similarly, a wide difference in the infection rate of internal fixation devices exists, considering that higher infective risk occurs after the fixation of open fractures (up to 30%) compared to closed fractures (less than 2%) [1].

Infections associated with orthopaedic implants, especially prosthetic joint infections (PJIs), are commonly classified as early-onset, delay-onset, and late-onset infections, with infections occurring in the first 3 months, between 3 and 12–24 months, or after 24 months, respectively [1,4]. Causative pathogens consist of virulent microorganisms (e.g., *Staphylococcus aureus*, *Enterococcus* spp., Gram-negative bacilli) in case of early-onset infections, while low-virulence pathogens, e.g., coagulase-negative staphylococci (CoNS), *Propionibacterium acnes*, are mostly involved in delayed- and late-onset infections [1,4]. Furthermore, it is important to recognize the resistance pattern of involved pathogens, considering that methicillin-resistant *S. aureus* (MRSA) strains may account for 25–50% of isolates [5]. Notably, *S. aureus* represent the leading causative pathogen in infections associated with orthopaedic implants, exhibiting remarkable virulence, particularly in patients with multiple comorbidities, and being involved in both early-onset and delayed- or late-onset cases [2]. Concomitant bacteraemia is retrieved in up to 60% of patients with infections associated with orthopaedic implants, with higher rates in patients treated with debridement with prosthesis retention than in patients with resection arthroplasty [2].

The wide variety of clinical scenarios encountered in the heterogeneous world of infections associated with orthopaedic implants makes the implementation of an optimal and standardized antimicrobial treatment in terms of drug choice, dosage regimen, and treatment duration challenging. While pathophysiological conditions exhibit a lower relevance compared to critically ill patients in the definition of the antimicrobial therapy puzzle [6], antibiotic bone penetration [7,8], anti-biofilm activity [9], long-term safety, and drug choice/dosage regimen favouring outpatient management (i.e., long-acting or oral agents), especially in the current COVID-19 era [10], play a major role in the chronic evolution of these infections (Figure 1).

Furthermore, the coordinated approach of a multidisciplinary task force including medical and surgical specialists (namely infectious disease consultants, clinical microbiologists, MD clinical pharmacologists, orthopaedic surgeons, intensive care physicians) could maximize treatment efficacy and improve clinical outcomes, as reported in other settings [11,12].

The Tuscany region has implemented an innovative organization called Rete AID (where A, I, and D refer to Antimicrobial Stewardship, Infection Control, and Diagnostic Stewardship, respectively) [13] for the systemic approach to fight microbiological threats. This organization can access a large information set provided by the collaborative network of clinical microbiology laboratories in the region, using the Regional Health Agency ARS (SMART network, Microbiological Surveillance and Antibiotic Resistance in Tuscany). In this context, a multidisciplinary task force, called Network Lab, including microbiologists, infectious disease specialists, orthopaedic surgeons, cardiac surgeons, and intensive care physicians, was implemented in order to develop a best practice model in the diagnosis and management of infections associated with implantable devices. In Tuscany, the rate of early-onset and delay-/late-onset infections associated with orthopaedic implants accounted for less than 1% and approximatively 2% of infections, respectively, while the rate of MRSA isolates decreased in recent years to below 30% [14].

Although several guidelines concerning the management of bone and joint infections have been issued in the last decade [15,16,17], a comprehensive overview focusing on the rational choice of available therapeutic alternatives from a microbiological and pharmacological standpoint is lacking [15].

This multidisciplinary opinion article aims to summarize evidence supporting the use of the different anti-staphylococcal agents in terms of microbiological and pharmacological optimization and to provide a useful guidance for clinicians in the management of patients affected by infections associated with orthopaedic implants.

## 2. Results

### 2.1. Issues in Microbiological Diagnosis of Infections Associated with Orthopaedic Implants

The microbiological diagnosis of orthopaedic implant-associated infections represents a crucial aspect of the management of these challenging conditions. As stated in the international guidelines [15], the detection of two or more positive cultures from intraoperative specimens, or a combination of preoperative aspiration and intraoperative cultures that yield the same organism (indistinguishable based on common laboratory tests including genus and species identification or common antibiogram) represents a definitive evidence of infection, providing acceptable sensitivity and specificity without requiring an impractical amount of tissue specimens to be processed by the laboratory [18]. Additionally, the growth of a virulent microorganism (e.g., *S. aureus*) in a single specimen of a tissue biopsy or synovial fluid may also be sufficient for the diagnosis of implant-associated infection [15]. Conversely, one of multiple tissue cultures or a single aspiration culture that yields an organism that is a common contaminant (e.g., CoNS, *Propionibacterium acnes*) should not necessarily be considered evidence of definite implant-associated infection and should be evaluated in the context of other available evidence [15].

Notably, microbiological diagnosis is also crucial for the identification of causing pathogens and their susceptibility, allowing the implementation of a targeted antibiotic therapy. At least 3, and optimally 5 or 6, periprosthetic intraoperative samples from the most suspicious areas of tissue as deemed by the orthopaedic surgeon should be obtained for aerobic and anaerobic culture for the optimal diagnosis of implant-associated infection, considering that the collection of fewer than 5–6 specimens leads to a decrease in the sensitivity of the culture as a diagnostic test [18,19]. The optimal duration of incubation of periprosthetic tissue specimens is currently unknown, but prolonged incubation of up to 14 days may help the isolation and identification of slow-growing low-virulent pathogens (e.g., *Propionibacterium* spp.) [20]. The implementation of novel processing techniques may also improve pathogen identification [21]. Sonication of the explanted prosthesis, with the aim to dislodge pathogens from the surface, followed by subsequent aerobic and anaerobic culture may improve the diagnostic sensitivity compared to traditional tissue cultures, increasing from 60.8% to 78.5% [19,20,21,22,23].

Novel molecular testing, e.g., multiplex polymerase chain reaction (PCR) performed on joint fluid aspirate may provide a rapid and accurate diagnosis of implant-associated infections, considering that higher diagnostic accuracy for the detection of PJIs and higher detection of low-virulence pathogens (e.g., CoNS, *Propionibacterium* spp.) compared to traditional cultures were reported [2,24,25].

### 2.2. Role of Bacterial Biofilm

The presence of biofilm plays a crucial role in the pathogenesis of orthopaedic implant-associated infections [2,9]. Both virulent bacteria (e.g., *S. aureus*) and opportunistic pathogens (e.g., CoNS) are able to form biofilm [9]. The formation and maturation of bacterial biofilm protects the pathogen from both the host immune response and antibiotics, making the treatment and eradication of this condition extremely challenging, considering that biofilm-embedded bacteria are 100–1000 times less susceptible to antibiotics [9]. Furthermore, biofilm-embedded bacteria are usually in a slow-growing or stationary state, leading to the development of bacterial subpopulations persisting in the biofilm in a dormant metabolic state that is highly resistant to antibiotics, known as small colony variants (SCVs) [26,27]. SCVs are characterized by unusual morphological appearance and biochemical reactions, often remaining undetected or misdiagnosed [27]. Additionally, they may exhibit an unstable phenotype, possibly reverting to the highly virulent and rapidly growing form [26,27].

An ideal anti-biofilm antibiotic should provide bactericidal activity, efficacy against bacteria in a stationary phase and biofilm-producing pathogens, and the ability to penetrate within biofilm as necessary requirements (Figure 2) [9].

In order to assess the anti-biofilm activity of a certain agent, classic antibiotic susceptibility tests providing the MIC required to define clinical breakpoints and PK/PD targets are inadequate [28]. Consequently, other pharmacodynamic parameters were developed to quantify antimicrobial activity against biofilm-producing pathogens, namely the minimal biofilm inhibitory concentration (MBIC), the minimal biofilm bactericidal concentration (MBBC), and the minimal biofilm eradication concentration (MBEC) [28]. However, the MBECs for common anti-staphylococcal agents are between 10- and 8000-fold higher than the MICs, thus making the choice of the best antibiotic strategy extremely challenging [9].

### 2.3. Real-World Use of Traditional and Novel Anti-Staphylococcal Agents for the Management of Infections Associated with Orthopaedic Implants

#### 2.3.1. Novel Lipoglycopeptide (Dalbavancin, Oritavancin, and Telavancin)

Several real-world experiences [29,30,31,32,33,34,35,36,37,38,39,40,41,42,43,44,45,46,47,48,49,50,51,52,53,54,55,56,57] document the efficacy of novel lipoglycopeptide, namely dalbavancin, oritavancin, and telavancin, in the management of infections associated with orthopaedic implants (Appendix A).

Currently, the evidence supports the use of dalbavancin in this scenario, with an overall clinical success rate above 75–80% [10,29]. Specifically, nine prospective and retrospective studies [30,31,32,33,34,35,36,37,38], one case series [39], and four case reports [40,41,42,43] assessed the efficacy and safety of dalbavancin for the treatment of PJIs and implant-associated infections. Morata et al. [30] reported the largest sample size, retrospectively studying 64 patients with bone and joint infections, of which 45 and 19 were affected by implant-associated infections and osteomyelitis/PJIs, respectively. *Staphylococcus epidermidis* was isolated in almost half of the cases. A clinical success or improvement was respectively reported in 97.7% and 89.5% of subjects with implant-associated infections and osteomyelitis/PJIs. Notably, a wide heterogeneity in the dalbavancin dosing schedule and treatment duration was noted. Overall, the rate of adverse events (AEs) ranged from 2% to 13%, most of which were classified as non-serious. 

Real-world evidence documenting the efficacy of oritavancin is currently limited to a retrospective study [44] and a case series [45] including only four patients affected by infections associated with orthopaedic implants. Overall, a clinical cure and/or improvement was documented in 75% of cases, while no AEs were reported. 

A retrospective study [46] and a case series [47] assessed the efficacy and safety of telavancin for the management of infections associated with orthopaedic implants. Notably, Sims et al. [46] retrospectively studies 57 patients, of which 30 were affected by osteomyelitis with prosthetic materials and 27 were affected by PJIs. *S. aureus* accounted for more than half of cases. Overall, the clinical cure was 75% and 84.6% in the osteomyelitis and PJIs group, respectively. AEs were reported in approximatively 20% of cases, of which 2.1% was classified as serious.

In several in vitro experiences, all of the three novel lipoglycopeptides showed high anti-biofilm activity against *Staphylococci* and *Enterococci* clinical isolates. Both lower minimum biofilm bactericidal concentrations (MBBCs) and minimal biofilm eradication concentrations (MBECs) were found when compared to treatment with vancomycin, linezolid, tedizolid, and cloxacillin [48,57]. Additionally, an additive or synergistic anti-biofilm activity was found for dalbavancin and oritavancin in combination with rifampicin [53,56].

Notably, both dalbavancin and oritavancin exhibited peculiar pharmacokinetic properties, consisting in a long half-life (higher than 7–14 days) and high bone penetration [58], thus allowing for favourable dosing regimens in terms of the outpatient management of infections associated with orthopedic implants. Specifically, preclinical studies showed that for both dalbavancin and oritavancin, bone concentrations were well above MIC_90_ of susceptible pathogens for up to 7–14 days [59,60]. Conversely, telavancin exhibited a short half-life (7–9 h); thus, daily administration is required [58]. Furthermore, studies assessing telavancin bone penetration is lacking. 

#### 2.3.2. Novel Anti-Staphylococcal Cephalosporins (Ceftaroline and Ceftobiprole)

Real-world evidence concerning the use of novel anti-MRSA cephalosporins, namely ceftaroline and ceftobiprole, in the management of infections associated with orthopaedic implants is currently limited (Appendix A).

A retrospective cohort study including only 19 patients, of which 16 were affected by implant-associated infections, assessed the efficacy of ceftaroline in this scenario [61]. Overall, a clinical cure was achieved in 68.4% of cases at end of treatment, but was maintained at one year in only approximatively 10% of patients. Furthermore, a remarkable rate of AEs was reported (more than 20%). Ceftaroline exhibited promising in vitro activity against *S. aureus* and *Staphylococcus epidermidis* strains retrieved in PJIs, with MIC_90/50_ below 0.5 mg/L [62]. Notably, ceftaroline exerted a bactericidal activity against both MSSA and MRSA biofilms, although a lack of complete biofilm eradication was found [63]. An additive and/or synergic effect with daptomycin and rifampicin was reported in preclinical/in vitro models of infections associated with medical devices, including PJIs [64,65].

Real-world evidence concerning ceftobiprole administration for the management of bone and joint infections is limited to a single retrospective study including nine patients [66]. The findings were promising in terms of a clinical cure (77.8%) and microbiological eradication rate (88.9%). Notably, no AEs were reported. Furthermore, preclinical/in vitro models showed an effective anti-biofilm activity and effective bone penetration compared to vancomycin or linezolid [67,68].

#### 2.3.3. Daptomycin

Three comparative studies [69,70,71], one prospective observational study [72], five retrospective cohort studies [73,74,75,76,77], and two case reports [78,79] investigated the efficacy and safety of daptomycin in the management of infections associated with orthopaedic implants, for a total of 277 included patients (Appendix A).

In a randomized controlled trial, Byren et al. [69] compared 49 patients affected by PJIs managed with two-stage revision receiving daptomycin 6–8 mg/kg/day with 26 patients receiving vancomycin, teicoplanin, or semisynthetic penicillins. Higher clinical cure (59.6% vs. 38.1%) and microbiological eradication (51.1% vs. 38.1%) rates were found in patients treated with daptomycin compared to other regimens. No difference in terms of AE rate leading to treatment discontinuation was reported (12.2% vs. 16.0%). In a retrospective comparative study including 341 patients affected by PJIs, Carli et al. [70] found that daptomycin use (*n* = 77) was not associated with better clinical outcome in patients undergoing either DAIR (OR 1.70; 95% CI 0.62–4.65) or two-stage exchange (OR 0.58; 95% CI 0.27–1.26) treatment when compared to other regimens. In a retrospective matched case-control study, Joseph et al. [71] reported no difference in either clinical cure (85% vs. 90%; *p* = 0.63) or relapse rate (10% vs. 10%; *p* = 0.99) in 20 patients receiving daptomycin for the management of PJIs compared to 20 cases treated with vancomycin. Notably, although no significant difference was reported in overall AE rate (20% vs. 30%; *p* = 0.47), cases receiving vancomycin experienced a higher rate of discontinuation due to severe AEs (25% vs. 0%; *p* = 0.02).

Overall, a remarkable clinical cure rate, ranging from 50.0% to 100.0%, was found in different retrospective studies and case reports [72,73,74,75,76,77,78,79] including patients affected by PJIs, mainly due to MRSA or MRSE, and managed with daptomycin alone or in combination with rifampicin. AEs were reported in up to 20% of patients. 

Notably, a phase I study [80] including 16 arthroplasty patients found a mean daptomycin bone penetration rate of 14.1 ± 11.9% after a dose of 8 mg/kg. Bone concentrations were well above MIC_90_ of pathogens susceptible to daptomycin.

Both in vitro and an animal foreign-body infection models [81,82] found promising results with the association of high-dose daptomycin with rifampicin or ceftaroline in biofilm-producing infections caused by MRSA or VRE. However, Molina Manso et al. [83] reported that daptomycin was not effective against biofilm produced by *S. aureus* and *Staphylococcus epidermidis* isolates in PJIs, with minimum biofilm eradication concentrations (MBECs) significantly above the minimum inhibitory concentrations (>1024 mg/L).

#### 2.3.4. Linezolid and Tedizolid

Four prospective observational studies [84,85,86,87], two retrospective matched case-control studies [88,89], and six retrospective cohort studies [90,91,92,93,94,95] investigated the efficacy and the safety of linezolid in the management of infections associated with orthopaedic implants, for a total of 365 included patients (Appendix A).

The overall clinical cure or remission rate was 79.7%, ranging from 68.2% to 100.0%. Notably, the overall AE rate in patients receiving long-term treatment with linezolid was remarkable, and AEs leading to linezolid discontinuation was reported in up to 14.3% of patients. Furthermore, no significant difference in clinical outcome was reported in patients affected by PJIs managed with DAIR receiving linezolid in monotherapy or in combination with rifampicin [90]. Similarly, no difference in clinical remission was found in 18 patients affected by orthopaedic implant-associated infections (of which 11 were PJIs) receiving linezolid in association with rifampicin, compared to 18 matched patients treated with cotrimoxazole plus rifampicin (89.3% vs. 78.6%; *p* = 0.47) [88].

The largest real-world experience was reported in the prospective observational study by Soriano et al. [84], in which 85 patients affected by orthopaedic device infections (69 PJIs, mostly due to MRSA or MRSE) receiving linezolid 600 mg q12h for a median of 60 days. The overall clinical remission rate was 68.2%, with 10.5% of patients experiencing at least one AE.

Real-world evidence concerning the use of tedizolid for the management of orthopaedic device-associated infections is limited to a retrospective multicenter study [96] and a single case report [97] (Appendix A). Benavent et al. [96] retrospectively studied 51 cases of bone and joint infections treated with tedizolid at a dosage of 200 mg/day for a median of 29 days. Coagulase-negative Staphylococci were isolated in approximatively half of the cases. A total of 29 orthopaedic implant-associated infections were included, 17 of which were PJIs, with a clinical cure rate of 83% (76% in patients affected by PJIs), while the overall AE rate was limited to 6% of patients (only mild AEs were reported).

Notably, in preclinical models, both linezolid and tedizolid were ineffective in eradicating mature biofilm formed by MRSA and MSSA, although a certain activity in preventing biofilm formation was noted, particularly for tedizolid [98,99]. 

Two different phase I studies [100,101] showed that linezolid at a dosage of 600 mg achieved effective bone and synovial concentrations that were at least double the MIC_90_ for both *Staphylococci* and *Streptococci*, in both patients undergoing joint replacement or affected by orthopaedic implant-associated infections.

#### 2.3.5. Vancomycin and Teicoplanin

Several guidelines recommended vancomycin as a first-line therapy in the management of prosthetic joint infections caused by oxacillin-resistant *Staphylococci* or penicillin-resistant *Enterococci* [15,17]. Different studies assessed the efficacy and the safety of novel agents (namely daptomycin and dalbavancin) compared to vancomycin in the management of orthopaedic implant-associated infections. As previously mentioned, no significant difference in the clinical outcome was found between daptomycin and vancomycin in the management of patients undergoing both DAIR or two-stage exchange [69,70,71], although a significantly lower occurrence of therapy discontinuation due to AEs was found for daptomycin [71]. Similarly, no difference in clinical cure at 1 year was found in 80 patients affected by osteomyelitis when randomized to dalbavancin or vancomycin [102], as well as in a retrospective case-control study including 215 patients (of which 102 were osteoarticular infections) receiving dalbavancin or vancomycin [103]. However, a significantly lower rate of AEs was reported in patients managed with dalbavancin compared to vancomycin [103]. No studies comparing vancomycin and linezolid were found in this scenario.

Notably, vancomycin showed poor activity in both preventing biofilm formation and in eradicating mature biofilm of the *staphylococci* and *enterococci* strains in different in vitro models when compared to novel lipoglycopeptides, with MBIC_90_ and MBBC_90_ higher than 1024 mg/L [48,51,55,57].

Phase I studies [104,105] assessing the bone penetration of vancomycin in patients affected by osteomyelitis or undergoing prosthetic joint replacement found a limited penetration in both cancellous (bone/serum ratio of 0.21) and cortical bone (mean bone/serum ratio of 0.3 ± 0.12). Moreover, Bue et al. [105] found a weaker bone penetration pattern in patients undergoing prosthetic joint replacement, with cancellous and cortical bone concentrations well below the MIC_50_ and MIC_90_ for *Staphylococci*, *Streptococci*, and *Enterococci*.

Teicoplanin was recommended as a first-line therapy for the management of orthopaedic implant-associated infections caused by methicillin-resistant *Staphylococci*, and as an alternative agent for multi-susceptible *Streptococci* and *Enterococci* in case of hypersensitivity to penicillin [106]. Peeters et al. [107] assessed 65 patients affected by bone and joint infections caused by *S. aureus* (17% MRSA) treated with teicoplanin alone or in combination therapy (fluoroquinolones, 45%, and rifampicin, 25%), of which 69% were classified as device-related infections. Teicoplanin was administered at a median dosage of 5.7 mg/kg/day after a loading dose consisting of five injections 12 h apart. Overall, clinical failure was found in 41.5% of patients after a median follow-up of 91 weeks, while AEs were reported in 10% of cases. Pavoni et al. [108] retrospectively studied 34 patients affected by PJIs, of which 22 received teicoplanin (mainly in combination with fluoroquinolones or rifampicin). Clinical improvement in the absence of relapse was documented in 61.9% of cases. No AEs were reported.

Interestingly, Claessens et al. [109] found that both teicoplanin and vancomycin were not effective in eradicating *S. epidermidis* biofilms, although the combination therapy with rifampicin improved the killing efficacy in vitro. Notably, different phase I studies found a teicoplanin bone/serum ratio ranging from 0.15 to 0.85 [110,111,112]. Furthermore, Lazzarini et al. [113] found a bone average concentration ranging from 0.9 to 2.94 mg/L in five patients undergoing total knee arthroplasty receiving 800 mg intravenous teicoplanin 2.5 h before surgery.

#### 2.3.6. Rifampicin

Several in vitro, animal, and clinical studies have documented the benefit of rifampicin in patients with staphylococcal orthopaedic implant-associated infections, as previously summarized in a recent review [114]. Furthermore, both in retrospective and prospective comparative studies, a significantly higher clinical cure rate was found in patients affected by staphylococcal PJIs undergoing DAIR and managed with rifampicin when compared to antibiotic regimens without rifampicin (80%-95% vs. 14%-68%) [114,115]. Notably, a randomized controlled trial [116] found a significantly greater clinical cure rate in 12 patients affected by staphylococcal PJIs treated with ciprofloxacin plus rifampicin when compared to 12 receiving ciprofloxacin alone (100% vs. 58%; *p* = 0.02). AEs were found in five cases receiving combination therapy with rifampicin and in one case treated with ciprofloxacin alone. Conversely, in a recent multicenter randomized controlled trial, Karlsen et al. [117] reported no significant differences in the remission rate between 23 patients randomized to combination therapy with rifampicin compared to 25 cases receiving standard antimicrobial therapy (cloxacillin or vancomycin) alone (74% vs. 72%; *p* = 0.88), although several issues with study design and conduction emerged [115].

As previously reported [53,55,56,81,82,99,114,115,118,119], preclinical evidence demonstrated the effective anti-biofilm activity of rifampicin in staphylococcal bone and joint infection models in association with dalbavancin, oritavancin, daptomycin, linezolid, fosfomycin, and fluoroquinolones.

In phase I studies, a rifampicin bone/serum ratio of 0.2–0.5 was found, with cortical and cancellous bone concentrations well above the MIC_50/90_ for *S. aureus* [7,120,121].

#### 2.3.7. Fluoroquinolones

Fluoroquinolones (i.e., levofloxacin, moxifloxacin, or ciprofloxacin), in association with rifampicin, are recommended as a first-line therapy for extended suppression (3–6 months) of staphylococcal PJIs [15]. A summary of the studies investigating combination therapy between fluoroquinolones and rifampicin in this scenario is provided in Appendix A.

Lora-Tamayo et al. [122] randomized 63 patients affected by staphylococcal PJIs managed with DAIR to receive a combination therapy of levofloxacin plus rifampicin for 8 weeks (short duration; 33 patients) or 3–6 months (long duration; 30 patients). After a median follow-up of 540 days, no difference in clinical cure rate was found between the two regimens in both intention-to-treat (58% vs. 73%; difference—15.7%, 95% CI −39.2% to 7.8%) and per-protocol analysis (95% vs. 91.7%; difference—3.3%, 95% CI −11.7% to 18.3%). Overall, AEs were reported in 15.7% of patients. Nguyen et al. retrospectively assessed 154 patients affected by PJIs (48.1% caused by CoNS) treated with a combination therapy of levofloxacin and rifampicin. After a mean follow-up of 55.6 months, an overall clinical cure rate of 82.5% was reported, although the occurrence of AEs was remarkable (31.2% and 8.4% attributable to rifampicin and levofloxacin, respectively) [123]. In a retrospective case-control study, Wouthuyzen-Bakker et al. [124] found no significant difference in clinical cure rate in 40 patients affected by PJIs due to MSSA managed with DAIR and treated with levofloxacin plus rifampicin when compared to 18 subjects receiving moxifloxacin plus rifampicin (89% vs. 87.5%; *p* = 0.89). Similarly, Fily et al. [125] reported a high clinical cure rate (78.3%) in 23 patients affected by non-staphylococcus PJIs treated with moxifloxacin plus rifampicin, although at least one AE occurred in 30.4% of cases.

Interestingly, conflicting evidence emerged from preclinical studies. While different models confirmed the high efficacy of the rifampicin-levofloxacin combination against biofilm-embedded bacteria, while also preventing the selection of resistant mutants that was observed with rifampin alone [99,119], an in vitro study including ten MSSA strains derived from PJIs showed that the addition of rifampicin to levofloxacin did not improve its performance. Additionally, an increase in small colony variants was observed in the presence of rifampicin [126].

Notably, high levofloxacin penetration was found in patients undergoing bone surgery, with a mean bone/serum ratio of approximatively 50% and 100% for cortical and cancellous bone, respectively [127,128]. Mean cortical and cancellous bone concentrations were well above the MIC_90_ for *S. aureus* and *Streptococci*. Similarly, a mean bone/serum ratio of approximatively 100% was found for moxifloxacin after a single administration of 400 mg before joint replacement surgery, with both cortical and cancellous bone concentrations well above the MIC_90_ for MSSA [120,129]. Ciprofloxacin bone penetration was not directly assessed in patients undergoing joint replacement surgery or affected by bone and joint infections. However, data collected during other surgical procedures documented the achievement of effective ciprofloxacin bone concentrations after a dosage of 400 mg iv or 750 mg oral, exceeding the clinical breakpoints of Gram-positive and -negative pathogens commonly retrieved in PJIs [7].

#### 2.3.8. Other Anti-Staphylococcal Agents (Fosfomycin, Cotrimoxazole, Tetracyclines, Clindamycin)

Fosfomycin, cotrimoxazole, tetracyclines, and clindamycin are recommended as alternative agents for the management of orthopaedic implant-associated infections in acute or chronic long-term scenarios [15].

Fosfomycin as part of a combination antimicrobial therapy at a dosage of 4–24 g/day was found to be highly effective in the management of bone and joint infections mainly caused by *S. aureus*, with an overall clinical cure rate of 82.2% and a limited occurrence of AEs [130]. Notably, a remarkable fosfomycin bone/serum ratio (0.43 ± 0.04) was found in nine patients with diabetic foot infections receiving fosfomycin 100 mg/kg/day [131].

Several results documented the efficacy of cotrimoxazole, tetracyclines (i.e., doxycycline and minocycline), and clindamycin for suppressive prolonged therapy in PJIs, as also recommended by international guidelines [15]. In a retrospective cohort study including 78 patients affected by PJIs (72.1% *Staphylococcus* spp.) receiving long-term suppressive therapy with doxycycline (92%) or minocycline (8%), Pradier et al. [132] found a clinical failure rate of 28.2% after a mean follow-up of 1020 days. In three cases, a documented acquisition of tetracycline resistance in the index pathogen was documented, while AEs were reported in 18% of cases, of which 8% lead to tetracycline discontinuation [132]. In another retrospective study including 39 patients affected by PJIs (28.2% MRSA; 84.6% undergoing DAIR) receiving doxycycline-based suppressive antimicrobial therapy, clinical cure was reported in 74.4% of cases (20.5% relapse; 5.1% superinfections) [133]. AEs were reported in six patients (15.4%), leading to doxycycline discontinuation in two of them. In a multicenter retrospective study including 302 patients affected by PJIs (63.6% *Staphylococcus* spp.) receiving long-term suppressive antimicrobial therapy (tetracycline and cotrimoxazole in 39.7% and 35.4% of the cases, respectively) for a median time of 36.5 months, a clinical success rate of 58.6% was found. AEs leading to therapy discontinuation occurred in 5.6% of patients [134]. Twenty-three patients affected by PJIs after hip replacement and receiving long-term suppressive therapy (60.9% doxycycline; 26.1% cotrimoxazole) were retrospectively studied [135]. After a median follow-up of 33 months, clinical success was documented in 56.5% of patients, with relapse occurring in 7 cases. A total of 26.1% of patients experienced AEs during long-term suppressive antimicrobial therapy, leading to treatment discontinuation in two cases [135]. Twenty-one patients affected by PJIs mainly caused by *Staphylococcus* spp. and receiving suppressive antimicrobial therapy with minocycline (67%) or clindamycin (83%) were retrospectively assessed [136]. Clinical success was found in 67% of patients, while AEs requiring a switch of antibiotic treatment or dosage adjustment were reported in 43% of cases. Cotrimoxazole, mainly in association with fluoroquinolones or rifampicin, was found to be an effective salvage therapy in 51 patients affected by bone and joint infections (76.1% device-associated), with a favourable outcome in 78.4% of cases at 90 days [137]. Courjon et al. [138] retrospectively assessed 196 patients affected by bone and joint infections (41% device-associated; 81% *Staphylococcus* spp.) and treated with clindamycin (in combination therapy with fluoroquinolones or rifampicin in 31% and 27% of cases, respectively). After a mean follow-up of 28 months, clinical remission was found in 57% of patients (83% of those assessable), while AEs were recorded in only nine patients. Notably, no difference in terms of efficacy and safety was found in 18 patients receiving combination therapy with cotrimoxazole and rifampicin when compared to 18 cases treated with linezolid plus rifampicin as a prolonged oral therapy for orthopaedic implant-associated infections [88].

In vitro evidence showed poor antibiofilm activity of cotrimoxazole, clindamycin, fosfomycin, and eravacycline against *S. aureus* and *S. epidermidis* strains isolated from PJIs [83,139]. Interestingly, effective bone concentrations were found in different clinical studies for doxycycline, cotrimoxazole, and clindamycin, exceeding MIC_50/90_ of *Staphylococcus* isolates retrieved in orthopaedic implant-associated infections [7,120]. The bone/serum concentration ratio was 0.21–0.45 for clindamycin and 0.2–0.3 for cotrimoxazole [7,120].

## 3. Discussion

The choice of the best antibiotic strategy for the management of patients affected by orthopaedic implant-associated infections currently remains extremely challenging, requiring a careful assessment of the PK/PD features of the different alternatives. The ideal antibiotic should provide a well-documented efficacy in real-world experiences for the treatment of implant-associated infections, showing both good bone penetration and adequate anti-biofilm activity. Furthermore, considering that these infections commonly required a long duration of treatment, both safety/tolerability and feasibility for outpatient administration represent important issues (Figure 3).

Novel long-acting lipoglycopeptides (namely dalbavancin and oritavancin) showed optimal bone penetration and good anti-biofilm activity coupled with an excellent safety profile and long-term tolerability [10,140]. Real-world evidence demonstrated optimal efficacy of dalbavancin in this scenario, with an overall clinical cure rate exceeding 80%, thus making it a potential first-line choice for orthopaedic implant-associated infections. Although real-world evidence assessing the efficacy of oritavancin in implant-associated infections is currently limited, a promising role and a growing use could be supposed for this antibiotic. By virtue of their long half-life, both dalbavancin and oritavancin exhibit an ideal profile for the outpatient management of patients affected by implant-associated infections, considering that a single administration of 1200 mg for oritavancin, or a double administration of 1500 mg for dalbavancin one week apart is sufficient to achieve optimal concentrations for up to five weeks [140,141]. Therapeutic drug monitoring (TDM) may play a crucial role in providing real-time feedback for the estimated duration of the optimal treatment of staphylococcal orthopaedic infections with dalbavancin in each single patient [142].

As regard other novel anti-staphylococcal agents, real-world evidence assessing the efficacy of telavancin, ceftaroline, and ceftobiprole in orthopaedic implant-associated infections is still limited. Considering that these agents require a daily intravenous administration, their use in an outpatient regimen could be extremely challenging. Although telavancin, ceftaroline, and ceftobiprole may exhibit a promising anti-biofilm activity, further investigation is required. Additionally, both bone penetration and long-term safety of these agents remain an open issue.

The real-world use and the efficacy of daptomycin and linezolid in implant-associated infections is well-established. Both agents show excellent bone penetration, although their anti-biofilm activity is poor; thus, they should be used in combination therapy with rifampicin or other anti-biofilm agents. Considering its excellent oral bioavailability, linezolid is widely used as an outpatient therapeutic regimen for staphylococcal prosthetic infections. However, long-term safety represents a major issue, given that AEs (particularly myelotoxicity and neurotoxicity) occur in most of cases. TDM may prove to be remarkably helpful for improving safety outcomes in patients requiring long-term treatment with linezolid [143,144,145]. Although preliminary evidence showed a better safety profile and anti-biofilm activity of tedizolid compared to linezolid, further investigation is required [146]. Additionally, real-world evidence assessing the efficacy of tedizolid in prosthetic infections is very limited, while bone penetration remains an open issue. However, considering the physicochemical and PK features tedizolid, it could be suggested that no significant difference exists when compared to linezolid in terms of bone penetration.

Although vancomycin and teicoplanin are widely used for the management of orthopaedic prosthetic infections caused by MRSA, the lack of antibiofilm-activity coupled with poor bone penetration (particularly for vancomycin) represent major concerns. Furthermore, the need for daily intravenous administration makes vancomycin an unappealing alternative for outpatient management. Conversely, according to its long half-life, evidence demonstrates the feasibility of administering teicoplanin as an effective strategy for the outpatient management of patients affected by orthopaedic implant-associated infections [147,148].

Rifampicin currently represents a cornerstone in the management of staphylococcal orthopaedic implant-associated infections, considering its excellent anti-biofilm activity, good bone penetration, and feasibility for outpatient management. Rifampicin is commonly used as a companion drug for the management of PJIs caused by MSSA or MRSA managed with DAIR or one-stage exchange, as well as in a combination regimen with fluoroquinolones, tetracyclines, cotrimoxazole, or clindamycin for chronic oral antimicrobial suppression [15]. Long-term safety and tolerability represent a major concern for rifampicin and fluoroquinolones [149], and to a lesser degree, for cotrimoxazole and tetracyclines.

Although a longer follow-up duration was mainly implemented in studies assessing the role of daptomycin, linezolid, and fluoroquinolones when compared to novel lipoglycopeptides, this aspect had no impact on the feasibility of treatments, being essentially related to the different outcome definitions used in retrospective observational studies and case series.

Notably, while methods for assessing the MIC are well standardized, these are not fully established for determining the antibiofilm activity of a specific antibiotic. However, the methods and the criteria selected for the determination of the MBIC, MBBC, and MBEC values for the same agents are similar and comparable, thus having no impact on the assessment of the antibiofilm activity of the different antibiotics.

Local delivery of antibiotics may represent an attractive and complementary therapeutic strategy for the treatment of biofilm-associated orthopaedic implant-associated *Staphylococcal* infections [150,151]. In this scenario, the implant of antibiotic-loaded carrier material may ensure the achievement of high antibiotic concentrations locally, thus overcoming the poor vascularity and biofilm formation potentially involved in the failure of a systemic antibiotic approach [151]. Although polymethylmethacrylate (PMMA) was initially used as an antibiotic carrier, its undesirable features (the need for subsequent removal, potential biofilm formation on surface) caused it to be replaced by other absorbable materials (calcium sulphate beads, hydrogel) [151,152]. Vancomycin, gentamycin, and tobramycin are the most broadly used agents for local delivery [151,152,153]. The delivery of local rifampicin has raised concerns about the development of rapid resistance when used as a single agent [150]. Conversely, the addition of fosfomycin or dalbavancin to antibiotic-loaded bone cement in orthopaedic implant-associated *Staphylococcal* infections showed high concentrations, with promising data in experimental biofilm models [150,154]. Although the use of local antibiotic delivery seems to be promising, there is still a lack of level 1 evidence to demonstrate the efficacy of this strategy [150]. Currently, the evidence supports the implementation of local antibiotic treatments mainly in the prophylaxis of periprosthetic or fracture-related infections, while therapeutic use remains an unmet clinical need [152,155]. 

Considering the extreme challenge of the treatment of orthopaedic implant-associated infections, the implementation of a multidisciplinary taskforce managing each stage of these conditions is essential in order to improve clinical outcome. Specifically, clinical microbiologists play a crucial role in the diagnosis of orthopaedic prosthetic infections, providing the identification of causative pathogens and their susceptibility, thus allowing the implementation of the best targeted antibiotic therapy. In this scenario, the implementation and the development a “network lab,” in which a prompt and close collaboration between microbiologists and clinicians takes place, as performed in the Tuscany region, could represent a model of best practice in the management of implant-associated infections.

Limitations of our study have to be addressed. No systematic search was performed; thus, we cannot exclude that some articles are missing. However, the assessment of the predefined major determinants was adequately supported for each antibiotic. Most of the included studies were conducted retrospectively, with a limited sample size. Furthermore, thresholds for assessing the predefined major determinants were arbitrarily defined according to the specific long-standing experience and expertise of each single member. Finally, the methods selected for the determination of antibiofilm activity are not standardized as are those commonly used for MIC determination. However, the criteria used in the different studies are comparable, thus providing reliable results for each agent in terms of specific antibiofilm activity.

## 4. Materials and Methods

A literature search was conducted on PubMed-MEDLINE (from inception until 19 December 2021) in order to retrieve studies investigating the use of traditional and novel anti-staphylococcal agents in the management of infections associated with orthopaedic implants. All antibiotics with documented activity against methicillin-resistant *Staphylococcus* spp. and cited in international guidelines performed by the IDSA for the diagnosis and management of prosthetic joint infections [15] were included. Additionally, other agents with documented evidence for the management of infections associated with orthopaedic implants caused by methicillin-resistant *Staphylococcus* spp. (namely dalbavancin, oritavancin, telavancin, ceftaroline, ceftobiprole, tedizolid, and fosfomycin) were also included. Agents with activity only against methicillin-resistant *Staphylococcus* spp. and cited in the IDSA guidelines [15] were excluded. 

A multidisciplinary task force composed of microbiologists, infectious disease specialists, orthopaedic surgeons, and intensive care physicians operating in the Tuscany region and characterized by specific expertise in the management of infections associated with orthopaedic implants identified different main topics focusing on unmet clinical needs in this scenario, namely diagnostic issues and the role of bacterial biofilm from a microbiological standpoint, and real-world evidence for the use of traditional and novel anti-staphylococcal agents, with a specific assessment of bone penetration, anti-biofilm activity, long-term safety, and feasibility for outpatient regimens. The definitive agreement for the identification and selection of each major determinant was reached by the multidisciplinary team after thorough discussion based on specific long-standing experiences and on the specific expertise of each single member in the management of infections associated with orthopaedic implants.

Specific thresholds were arbitrarily defined on the basis of the specific expertise of the different members in order to assess the performance of selected antibiotics for each of the five major determinants:(a)Efficacy in implant-associated infections: at least three studies including more than 70 patients and an overall positive clinical outcome of at least 70% for scoring the performance of the agent as optimal;(b)Bone penetration: existence of at least one preclinical/clinical study reporting the achievement of antibiotic bone concentrations above the MIC_50_/MIC_90_ for *Staphylococcus* spp. for scoring the performance of the agent as optimal;(c)Antibiofilm activity: existence of at least one in vitro study reporting the potential achievement of antibiotic concentrations above the MBIC for scoring the performance of the agent as optimal;(d)Long-term safety: an overall AEs rate below 20% in the included clinical studies for scoring the performance of the agent as optimal, while a proportion of the overall AEs above 40% identified agents with poor long-term safety;(e)Feasibility for outpatient management: the availability of oral formulations, and/or the possibility to perform once/twice-weekly administration, as well as feasibility for outpatient parenteral antibiotic therapy were considered for scoring the performance of the agent as optimal.

The following terms were searched on PubMed in combination: prosthetic joint infections, orthopaedic implant infections, bone infections, biofilm, bacterial biofilm, bone penetration, long-term, safety, dalbavancin, oritavancin, telavancin, daptomycin, linezolid, tedizolid, teicoplanin, ceftaroline, ceftobiprole, rifampicin, levofloxacin, fosfomycin, cotrimoxazole, minocycline, doxycycline, and tetracycline.

For each included study, the following information was extracted: (a) study author and year of publication; (b) study characteristics, including study design and sample size; (c) features of the patients, including site of infection, isolated pathogens, antibiotic therapy and duration, dosing schedule, and duration of follow-up; (d) types of outcome measurements, including rate of clinical success or improvement, clinical and/or microbiological failure rate, mortality rate, relapse rate, resistance development, and the overall proportion of adverse events.

The quality of the evidence was established according to a hierarchical scale of the study design, as reported in the evidence pyramid [156]: randomized controlled trials (RCTs); prospective observational studies; retrospective observational studies; case series; case reports; preclinical/in vitro studies. Clinical and preclinical pharmacokinetic or pharmacokinetic/pharmacodynamic (PK/PD) studies, as well as microbiological studies, were retrieved in order to support the choice of each selected anti-staphylococcal agent.

The consistency between the retrieved studies was also considered by assessing the concordance in clinical outcome (for clinical studies) and/or specific endpoints (for preclinical/in vitro studies) of the included studies at each level of the evidence pyramid. Only articles published in English were included, and the search was focused mainly on the last ten years in order to provide an up-to-date overview on the scientific evidence that may support the therapeutic choice.

## 5. Conclusions

The implementation of the best antibiotic strategy in patients affected by orthopaedic implant-associated infections remains extremely challenging, persisting as a matter of debate despite available guidelines. In this scenario, the existence of firm evidence establishing the efficacy of the different agents, stemming from well-conducted and possibly comparative studies, is crucial for guiding physicians in the choice of the best therapeutic strategy. Furthermore, the assessment of adequate bone penetration and anti-biofilm activity, coupled with long-term tolerability and feasibility for outpatient management, represent key issues for the treatment of orthopaedic implant-associated infections. Among available antibiotic treatments, novel long-acting lipoglycopeptide, particularly dalbavancin, alone or in combination with rifampicin, could represent the best antibiotic choice according to real-world evidence and PK/PD properties. The implementation of a multidisciplinary taskforce and of a close cooperation between microbiologists and clinicians is crucial for providing the best care in this scenario. The clinical interpretation of the TDM of the different antibiotics performed by a well-trained clinical pharmacologist could represent an added value for maximizing the clinical efficacy and minimizing toxicity in patients affected by orthopaedic implant-associated infections.

## Figures and Tables

**Figure 1 antibiotics-11-00406-f001:**
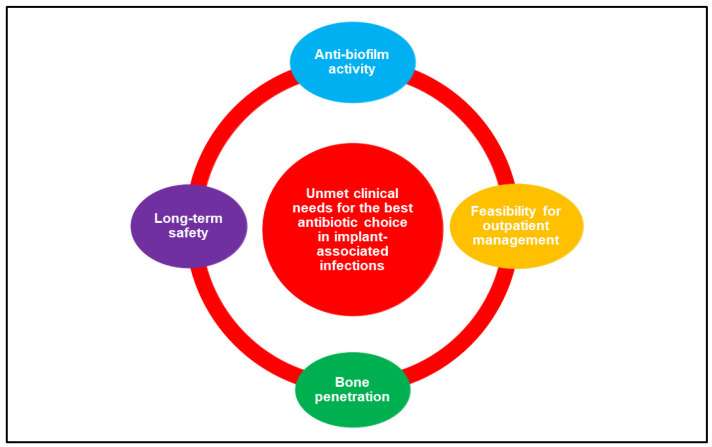
The major determinants involved in the choice of the best antibiotic strategy for the management of orthopaedic implant-associated infections.

**Figure 2 antibiotics-11-00406-f002:**
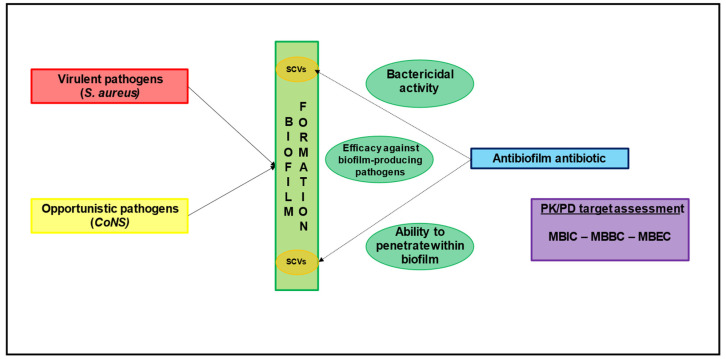
The role of biofilm formation and the major determinants involved in the choice of antibiofilm agents. CoNS: coagulase-negative Staphylococci; MBBC: minimal biofilm bactericidal concentration; MBEC: minimal biofilm eradication concentration; MBIC: minimal biofilm inhibitory concentration; PK/PD: pharmacokinetic/pharmacodynamic SCV: small colony variants.

**Figure 3 antibiotics-11-00406-f003:**
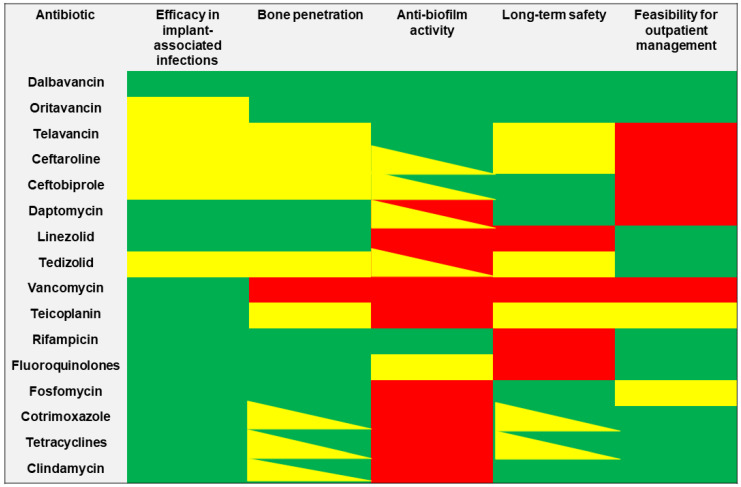
The features of selected anti-staphylococcal agents in relationship with major determinants required for the management of orthopaedic implant-associated infections. Green box: optimal activity/efficacy; yellow box: some concerns in activity/efficacy; red box: limited or absent activity/efficacy.

## Data Availability

All data are retrieved from publicly available papers.

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
