# Peer review of "Orthopaedic Implant-Associated Staphylococcal Infections: A Critical Reappraisal of Unmet Clinical Needs Associated with the Implementation of the Best Antibiotic Choice"

_antibiotics, 2022, doi:10.3390/antibiotics11030406_

Round 1

Reviewer 1 Report

This Perspectives conducting a systematic search on summarize evidence supporting the use of the different anti-staphylococcal agents in terms of microbiological and pharmacological optimization and to provide a useful guidance for clinicians in the management of patients affected by infections associated with orthopaedic implants. This review indicated that supporting the use of traditional and novel anti-staphylococcal agents in orthopaedic implant-associated infections were summarized, with a specific assessment of bone penetration, anti-biofilm activity, long-term safety, and feasibility for outpatient regimens. Overall, this manuscript is very interesting and well-written.

Some specific comments below: 

Title : The review focus on stphalococcal infection and therapy. So why modify the title for just stphalococcal infection? for example:

Orthopaedic implant-associated stphalococcal  infections: a critical reappraisal of unmet clinical needs associated with the implementation of the best antibiotic choice.

Introduction: Need to add some information about Orthopaedic implant and Staphylococcus aureus.

Line 57: First time to use Staphylococcus aureus, please after that use  S. aureus in all text.

Line 98: Please add refrance

Section 2.2 Role of bacterial biofilm: you need add figure or table to illustrated the inorfmation in this part. I believe figure much better and appealing.

Conclusion: The summary of your Conclusion is somewhat brief.

Author Response

Manuscript ID: antibiotics-1618389, entitled "Orthopaedic implant-associated infections: a critical reappraisal of unmet clinical needs associated with the implementation of the best antibiotic choice" by Gatti et al.

Dear Editor,

We would like to thank you for the opportunity to resubmit a revised version of this manuscript. We appreciated the reviewers’ constructive comments. All have been carefully considered and incorporated, where and whenever possible, in the revision.

Our point-by-point responses are provided below.

Q= QUERY; A= ANSWER

Reviewer #1

This Perspectives conducting a systematic search on summarize evidence supporting the use of the different anti-staphylococcal agents in terms of microbiological and pharmacological optimization and to provide a useful guidance for clinicians in the management of patients affected by infections associated with orthopaedic implants. This review indicated that supporting the use of traditional and novel anti-staphylococcal agents in orthopaedic implant-associated infections were summarized, with a specific assessment of bone penetration, anti-biofilm activity, long-term safety, and feasibility for outpatient regimens. Overall, this manuscript is very interesting and well-written.

We thank the reviewer for the appreciation.

Some specific comments below: 

Q1. Title : The review focus on stphalococcal infection and therapy. So why modify the title for just stphalococcal infection? for example:

Orthopaedic implant-associated stphalococcal  infections: a critical reappraisal of unmet clinical needs associated with the implementation of the best antibiotic choice.

A1. We thank the reviewer for this suggestion. We modified the title according to the proposal provided by the reviewer.

Q2. Introduction: Need to add some information about Orthopaedic implant and Staphylococcus aureus.

A2. We thank the reviewer for this comment. We added some information in the introduction concerning the role of Staphylococcal aureus in orthopaedic implant infections (Line 63-68).

Q3. Line 57: First time to use Staphylococcus aureus, please after that use  S. aureus in all text.

A3. We modified and used S. aureus instead of Staphylococcus aureus in all text as suggested.

Q4. Line 98: Please add reference

A4. We used the international guidelines performed by the IDSA for the diagnosis and management of prosthetic joint infections as reference, considering that several points addressed in our paper concerning the management of orthopaedic implant associated infections were indicated as research gaps by the IDSA guidelines (e.g., What is the efficacy of oral vs parenteral therapy, or oral step-down therapy as an alternative to prolonged parenteral therapy? What is the efficacy of rifampin combination therapy for staphylococcal PJI? What are alternatives to vancomycin for the management of infection with MRSA or coagulase- negative Staphylococcus? What is the role of chronic suppression, where is it indicated, and how much is adequate? Which agents are appropriate for suppression?).

Q5. Section 2.2 Role of bacterial biofilm: you need add figure or table to illustrated the inorfmation in this part. I believe figure much better and appealing.

A5. We thank the reviewer for this interesting suggestion. We added a figure summarizing the information concerning the role of bacterial biofilm.

Q6. Conclusion: The summary of your Conclusion is somewhat brief.

A6. We thank the reviewer for this comment. We have extended the conclusion of our article (Line 658-673).

Reviewer 2 Report

What were the inclusion/exclusion criteria for selecting the specific antibiotics described in this study? The criteria must be described in the materials and methods section.

The criteria for selecting and scoring the major determinants required to manage orthopedic implant-associated infections should be discussed. For example, why are antibiofilm activity, bone penetration, long-term safety considered, but other factors are not?

Figure 2: 1) what is the threshold for assessing the “optimal” vs. “some concerns,” and what is the impact of the lack of information? 2) what is the number of articles for each determinant/antibiotic box in the figure 2 matrix?

Most of the lipoglycopeptides follow-up was for 30-180 days, while others (i.e., daptomycin, linezolid, or fluoroquinolones) was for >2 years. Is the difference in the follow-up duration associated with the infection or the type of antibiotic? Is it relevant for the feasibility of the treatments? If so, it should be discussed in the manuscript.

The methods for assessing the MIC are well standardized (i.e., CLSI, EUCAST); however, the methods for determining the antibiofilm activity are not that much. Did the author compare the MBBC/MBIC/MBEC values with the procedures and criteria used in the different publications? Different methods may yield different results even for the same drugs/conditions. As the antibiofilm activity is one of the major features for determining the optimal treatments, it should be addressed.

Is there any correlation between the material composition of orthopedic implants and the infection’s occurrence and associated pathogens? If so, how does the composition influence the effectiveness of the antibiotic treatments?

Is the role of secondary infections considered in the treatment efficacy?

Tables: improve the format to increase readability

Does Ref 14 have any associated online information?

Author Response

Manuscript ID: antibiotics-1618389, entitled "Orthopaedic implant-associated infections: a critical reappraisal of unmet clinical needs associated with the implementation of the best antibiotic choice" by Gatti et al.

Dear Editor,

We would like to thank you for the opportunity to resubmit a revised version of this manuscript. We appreciated the reviewers’ constructive comments. All have been carefully considered and incorporated, where and whenever possible, in the revision.

Our point-by-point responses are provided below.

Q= QUERY; A= ANSWER

Reviewer #2

Q1. What were the inclusion/exclusion criteria for selecting the specific antibiotics described in this study? The criteria must be described in the materials and methods section.

A1. We thank the reviewer for this comment, which allow us to clarify an important methodological aspect of our paper. All antibiotics with documented activity against methicillin-resistant Staphylococcus spp and cited by international guidelines performed by the IDSA for the diagnosis and management of prosthetic joint infections were included in our study. Additionally, other agents with documented evidence for the management of infections associated with orthopaedic implants caused by methicillin-resistant Staphylococcus spp (namely dalbavancin, oritavancin, telavancin, ceftaroline, ceftobiprole, tedizolid, and fosfomycin) were also included. Agents with activity only against methicillin-resistant Staphylococcus spp and cited in the IDSA guidelines were instead excluded. We added this important point in the Materials and Methods section (Line 595-602).

Q2. The criteria for selecting and scoring the major determinants required to manage orthopedic implant-associated infections should be discussed. For example, why are antibiofilm activity, bone penetration, long-term safety considered, but other factors are not?

A2. We thank the reviewer for this comment, which allow us to better clarify this important point. Considering that our paper represents a perspective article, no specific and systematic criteria were applied for selecting and scoring the major determinants required to manage orthopedic implant-associated infections. Bone penetration, anti-biofilm activity, long-term safety, and feasibility for outpatient regimens were identified as major determinants by the different members of the multidisciplinary task force after reaching an agreement through discussion for each single determinant based on their specific long-standing experience and expertise in the management of these infections. We discussed this important point in the Materials and Methods section (Line 610-614).

Q3. Figure 2: 1) what is the threshold for assessing the “optimal” vs. “some concerns,” and what is the impact of the lack of information? 2) what is the number of articles for each determinant/antibiotic box in the figure 2 matrix?

A3. We thank the reviewer for these comments, allowing us to better clarify these important methodological points. Specific thresholds identified for assessing the “optimal” vs. “some concerns” were selected according to the specific determinant evaluated. For example, as regards the item “Efficacy in implant-associated infections”, only two case series including four patients exist for oritavancin, one retrospective study and one case report exist for tedizolid (overall 30 patients), only one retrospective study exists for ceftaroline (19 patients) and ceftobiprole (17 patients), while only one retrospective study and one case series (63 patients) exist for telavancin, thus justifying the assessment as “some concerns” compared to other agents, for which at least three large studies (e.g., 14 for dalbavancin, 11 for daptomycin, or 12 for linezolid) are available. For the item “Feasibility for outpatient management”, the availability of oral formulations, or the possibility to perform once-weekly (or longer) administration, as well as feasibility for OPAT were considered for scoring this specific issue. As regard the item “Bone penetration”, the existence of preclinical or clinical studies assessing the achievement of bone concentration above the MIC50/MIC90 for Staphylococcus spp was considered for scoring this point as “optimal”. For the item “Long-term safety”, an overall AEs rate below 20% in the different included studies was identified as the specific threshold for scoring this point as “optimal”, while a proportion above 40% of AEs identified agents with poor long-term safety. Finally, as regard the item “Antibiofilm activity”, the potential achievement of antibiotic concentrations above the MBIC was identified as the specific threshold for scoring this point as “optimal”. Lack of information had no impact on the assessment of specific determinants, given that for each selected agent evidence are available for evaluating the performance in infections associated with orthopaedic implants. As regard the number of studies for each determinant/antibiotic box in Figure 2, these are all listed in Tables 1-5. We discussed this important point in the Materials and Methods section (Line 615-632).

Q4. Most of the lipoglycopeptides follow-up was for 30-180 days, while others (i.e., daptomycin, linezolid, or fluoroquinolones) was for >2 years. Is the difference in the follow-up duration associated with the infection or the type of antibiotic? Is it relevant for the feasibility of the treatments? If so, it should be discussed in the manuscript.

A4. We thank the reviewer for this interesting comment. As correctly noted, a longer follow-up was implemented for daptomycin, linezolid, or fluoroquinolones compared to lipoglycopeptides, although this aspect is not associated with the type of infections (generally infections were PJIs managed with DAIR or two-stage exchange for the different agents). Similarly, although dalbavancin or oritavancin may ensure an optimal anti-MRSA activity up to 6 weeks after the last dose in bone and joint infections (refer to Cojutti et al. doi: 10.1128/AAC.02260-20), thus representing the real advantage in terms of feasibility, the type of antibiotic is not involved in the difference of the observed follow-up duration. Considering that most evidence resulted from retrospective observational studies or case series with no standardized definition in terms of outcome or follow-up, it is expected that difference in follow-up duration was mainly related to study design. However, it is important to recognize that test of cure for prosthetic joint infections is usually assessed six weeks after the last antibiotic dose, thus a follow-up of 30-180 days as reported for studies investigating novel lipoglycopeptides may be justified and reliable. We briefly discussed this aspect in the discussion section (Line 569-575).

Q5. The methods for assessing the MIC are well standardized (i.e., CLSI, EUCAST); however, the methods for determining the antibiofilm activity are not that much. Did the author compare the MBBC/MBIC/MBEC values with the procedures and criteria used in the different publications? Different methods may yield different results even for the same drugs/conditions. As the antibiofilm activity is one of the major features for determining the optimal treatments, it should be addressed.

A5. We agree with this relevant comment. As correctly noted, while methods for assessing the MIC are well standardized, these are not fully established for determining antibiofilm activity. However, the methods and the criteria selected for the determination of the MBIC, MBBC, and MBEC values for the same agents are similar and comparable, thus having no impact on the assessment of antibiofilm activity of a specific antibiotic. We added a specific comment in the discussion section (Line 576-580).

Q6. Is there any correlation between the material composition of orthopedic implants and the infection’s occurrence and associated pathogens? If so, how does the composition influence the effectiveness of the antibiotic treatments?

A6. We thank the reviewer for this interesting comment. Recent evidence reported no correlation between the material composition of orthopedic implants and occurrence of infections and/or isolation of specific pathogens (refer to Tande et al. doi:10.1128/CMR.00111-13). Although older data suggest that metal-to-metal hinged-knee prostheses are more frequently infected than metal-to-plastic prostheses, large case-control and registry-based studies have found no difference between cemented and uncemented arthroplasties. Furthermore, data supporting the use of antimicrobial-loaded polymethylmethacrylate for prosthesis reimplantation in patients undergoing one-stage or two-stage arthroplasty exchanges are largely retrospective, and this approach does not appear to be a lower risk of reinfection when an uncemented revision is performed for PJI (refer to Tande et al. doi:10.1128/CMR.00111-13). Consequently, it is expected that the composition of orthopedic implants has no impact on the effectiveness of antibiotic treatments.

Q7. Is the role of secondary infections considered in the treatment efficacy?

A7. We thank the reviewer for this comment. The role of secondary infections was considered in clinical efficacy, given that relapse infections within follow-up of each study were considered as treatment failure.

Q8. Tables: improve the format to increase readability

A8. We thank the reviewer for this suggestion. We performed some small stylistic adjustments in order to improve readability of tables.

Q9. Does Ref 14 have any associated online information?

A9. We thank the reviewer for this comment. The reference n. 14 has an associated online information at the following link: https://www.ars.toscana.it/images/pubblicazioni/Collana_ARS/2020/Documento_ARS_107/Doc_112_SMART_2021_DEF.pdf. We added this information in the reference.